# Linking tundra vegetation, snow, soil temperature, and permafrost

Inge Grünberg[1], Evan J. Wilcox[2], Simon Zwieback[3], Philip Marsh[2], and Julia Boike[1,4]

[1]Permafrost Research, Alfred Wegener Institute Helmholtz Center for Polar and Marine Research, Potsdam, Germany
[2]Cold Regions Research Centre, Wilfrid Laurier University, Waterloo, ON, Canada
[3]Geophysical Institute, University of Alaska Fairbanks, Fairbanks, AK, United States of America
[4]Geography Department, Humboldt-Universität zu Berlin, Berlin, Germany

**Correspondence:** I. Grünberg (inge.gruenberg@awi.de)

**Abstract.** Connections between vegetation and soil thermal dynamics are critical for estimating the vulnerability of permafrost to thaw with continued climate warming and vegetation changes. The interplay of complex biophysical processes results in a highly heterogeneous soil temperature distribution on small spatial scales. Moreover, the link between topsoil temperature and active layer thickness remains poorly constrained. Sixty-eight temperature loggers were installed at $1$–$3\,\mathrm{cm}$ depth to record the distribution of topsoil temperatures at the Trail Valley Creek study site in the Northwestern Canadian Arctic. The measurements were distributed across six different vegetation types characteristic for this landscape. Two years of topsoil temperature data were analysed statistically to identify temporal and spatial characteristics and their relationship to vegetation, snow cover, and active layer thickness. The mean annual topsoil temperature varied between $-3.7\,^{\circ}\mathrm{C}$ and $0.1\,^{\circ}\mathrm{C}$ within $0.5\,\mathrm{km}^2$. The observed variation can, to a large degree, be explained by variation in snow cover. Differences in snow depth are strongly related with vegetation type and show complex associations with late-summer thaw depth. While cold winter soil temperature is associated with deep active layers in the following summer at lichen and dwarf shrub tundra, we observed the opposite beneath tall shrubs and tussocks. In contrast to winter observations, summer topsoil temperature is similar below all vegetation types with an average summer topsoil temperature difference of less than $1\,^{\circ}\mathrm{C}$. Moreover, there is no significant relationship between summer soil temperature or cumulative positive degree days and active layer thickness. Altogether, our results demonstrate the high spatial variability of topsoil temperature and active layer thickness even within specific vegetation types. Given that vegetation type defines the direction of the relationship between topsoil temperature and active layer thickness in winter and summer, estimates of permafrost vulnerability based on remote sensing or model results will need to incorporate complex local feedback mechanisms of vegetation change and permafrost thaw.

## 1 Introduction

Arctic ecosystems are changing rapidly, with widespread reports of air temperature increase (IPCC, 2013), decreasing area and duration of snow cover (AMAP, 2017), and warming and degrading permafrost (Biskaborn et al., 2019). Permafrost thaw depends on local influences on the transfer of heat into the ground including soil physical properties, hydrology, and vegetation. Permafrost ecosystems are undergoing rapid vegetation change with increasing shrub abundance, cover, and biomass in many regions (Tape et al., 2006; Myers-Smith et al., 2011; Sturm et al., 2001b; McManus et al., 2012; Lantz et al., 2013;

Frost and Epstein, 2014). Yet, permafrost models and remote-sensing-driven monitoring approaches are still limited in their representation of small–scale spatial variability of snow and vegetation (Langer et al., 2013; Zhang et al., 2014; Park et al., 2016).

The interaction of vegetation and active layer thickness is very complex if the energy balance and soil properties are considered simultaneously (Loranty et al., 2018). In winter, snow insulates the soil from the cold air temperature. The resulting
difference between the soil and air temperature is important for the ground thermal regime and thus permafrost temperatures (Zhang, 2005). Vegetation affects snow depth and density because tall shrubs trap snow (Pomeroy et al., 2006; Sweet et al., 2014); also some vegetation types preferentially grow in locations with deeper snow cover in order to be protected from cold air temperatures and profit from additional moisture in spring (Grippa et al., 2005; Sturm et al., 2005b). This association leads to warm soil temperature in winter below tall shrubs (Lantz et al., 2009; Frost et al., 2018) and in tussock tundra found in poorly
drained areas in our study region (with long freeze–back periods), but also occurs on mesic slopes elsewhere; in contrast, lichen tundra is usually associated with wind exposed hill tops and ridges which accumulate the least snow (Pomeroy et al., 1997; Burn and Kokelj, 2009).

Snow melt timing is considered to be one of the most important drivers of active layer thickness at the end of summer (Chapin et al., 2005; Wilcox et al., 2019). However, in certain areas the strong association may also be an artefact of the confounding
relationship with other variables which strongly affect the active layer thickness. For instance, wind-blown ridges with thin snow cover and hence early snowmelt also tend to store less organic matter, and are therefore less insulating (Michaelson et al., 1996) than more sheltered locations where more snow is deposited. Also, depressions and tall shrub sites that accumulate deep snow cover and exhibit late snowmelt are expected to be moister and to accumulate more peat and organic soil as compared to wind-exposed ridges (Walker et al., 2008; Pajunen et al., 2011). Therefore, the thicker moss and organic layer at tussock and
tall shrub sites may be the factor that helps to keep the active layer cooler and shallower, rather than the late snowmelt (Walker et al., 2008; Frost et al., 2013; Loranty et al., 2018).

Observational evidence indicates that increased shrub cover reduces summer soil temperatures and decreases seasonal permafrost thaw depths (Anisimov et al., 2002; Walker et al., 2003, 2008; Blok et al., 2010), but tall shrub expansion in the tundra warms soils on annual timescales (Loranty and Goetz, 2012). Summer surface temperature is reduced by tall vegetation through
radiation shading, rainfall interception, and evapotranspiration (Loranty and Goetz, 2012; Zwieback et al., 2019). Increased evapotranspiration may reduce summer soil warming as it is an energy sink and decreases soil moisture and thus soil thermal conductivity during summer (Fisher et al., 2016). While these factors lead to reduced summer warming underneath shrubs, the magnitude and even the sign of this association varies in space and time. To exploit vegetation cover as a proxy for permafrost soil temperatures in summer, detailed observations of soil temperature and vegetation cover are required. The influence of all
these factors on active layer thickness is complex and spatially variable, highlighting the difficulty of attributing active layer thickness trends to any one particular variable.

This study quantifies the complex relationship between vegetation cover, snow, topsoil temperature, and active layer thickness and explores the local seasonal variability of the four components. We hypothesise that in winter the dependence of topsoil temperatures on vegetation is largely shaped by the association of vegetation with snow depth, owing to the strong insulating

effect of snow. We also hypothesise that the timing of snowmelt is a dominant control on active layer thickness. Moreover, we expect that active layer thickness is reduced by tall vegetation through shading in summer. In the current study, we aim to improve our understanding of the feedback mechanisms in the complex permafrost–vegetation–atmosphere system across the four seasons. The results provide a basis for upscaling and modelling attempts and for assessing the potential of vegetation remote sensing for permafrost applications.

## 2   Methods

### 2.1   Field site, soil, and vegetation

The Trail Valley Creek research site is located at the tree line in the tundra-taiga transition zone $45\,\mathrm{km}$ north of Inuvik, Northwest Territories, Canada, east of the Mackenzie Delta ($133.499\,°\mathrm{W}$, $68.742\,°\mathrm{N}$, Figure 1a). The mean annual air temperature in

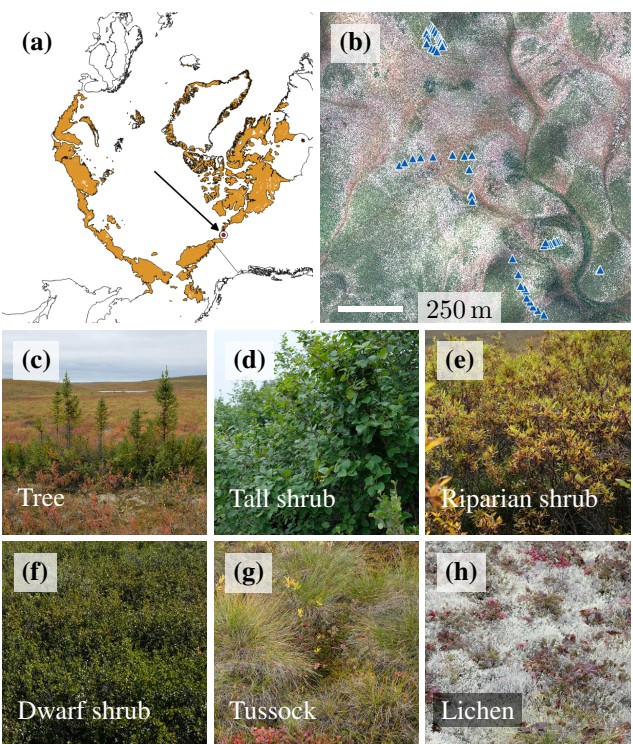

**Figure 1.** (a) Location of the Trail Valley Creek research station, Northwest Territories, Canada and Arctic tundra extent in orange colour (data from Walker et al., 2005), (b) airborne orthophoto (Polar 5 airplane, MACS camera, 08/2018) of the site including topsoil temperature measurement locations, (c–h) six vegetation types used in this study.

the 1999 – 2018 period was $-7.9\,°\mathrm{C}$ (Environment and Climate Change Canada, 2019). During this 20-year period, it rose by $1.1\,°\mathrm{C}$ per decade and the strongest warming trend was observed in May with an increase of $2.8\,°\mathrm{C}$ per decade (Environment

and Climate Change Canada, 2019). Although the site is only 70 km south of the Arctic Ocean, the climate is continental and summers can be quite warm (Figure B1). The study site is underlain by continuous permafrost 100–150 m thick (Marsh et al., 2008) and is characterised by an active layer 25–100 cm deep at the end of summer. The catchment is dominated by mineral-earth hummocks which have a 5 cm thick organic layer and are underlain by fine grained material composed of roughly one third sand, one third silt, and one third clay (Quinton et al., 2000). Between the hummocks is a several decimeters thick peat layer, known as the inter-hummock zone (Quinton et al., 2000). The permafrost is ice-rich and thus susceptible to warming and thawing (Burn and Kokelj, 2009).

Gently rolling hills structure the lowland landscape (Marsh et al., 2010) and provide habitats for different tundra vegetation types and, in favourable locations, forest patches. The vegetation can be divided into six main types (Figure 1c–h) based on the classification by Walker et al. (2005):

1) **Trees** can be found in river channels and on adjacent slopes as well as in isolated patches. While trees growing in the forest patches can reach 10 m height, trees in isolated patches are usually 0.5–2.5 m tall (Anders et al., 2018; Antonova et al., 2019). The tree species are white and black spruce (*Picea glauca* and *Picea mariana*) (Palmer et al., 2012). Forest patches cover roughly 2% of the landscape (Grünberg and Boike, 2019).

2) **Tall shrub** tundra is characterised by sparse patches of green alder (*Alnus alnobetula*) (Street et al., 2018); tall shrubs on hill slopes grow 3–5 m apart. While the alder shrubs are up to 2 m tall, the area between the shrubs is covered by shorter shrub species such as dwarf birch (*Betula glandulosa*) and by grasses and sedges. Tall shrubs (class S2 (low-shrub tundra) in Walker et al. (2005)) cover at least 11% of the wider Trail Valley Creek area (Grünberg and Boike, 2019).

3) **Riparian shrub** tundra can be found next to streams and at water tracks. Willow (*Salix*) species dominate these areas and grow up to 2.1 m tall. Additional shrub species include green alder and dwarf birch. Riparian shrubs (class S2 (low-shrub tundra) in Walker et al. (2005)) cover about 14% of the landscape (Grünberg and Boike, 2019).

4) **Dwarf shrub** tundra is one of the most abundant vegetation types growing on hill tops and slopes. In general, dwarf birch forms a dense canopy 20–50 cm high; in extreme cases it can reach 1 m. Dwarf birch is often complemented by shorter dwarf shrubs such as Labrador tea (*Ledum palustre*) and mountain cranberry (*Vaccinium vitis-idaea*), forbs (e.g. sweet coltsfoot, *Petasites frigidus*), graminoids, mosses, and lichen (Street et al., 2018). Dwarf shrubs (class S1 (erect dwarf-shrub tundra) in Walker et al. (2005)) cover roughly 24% of the landscape (Grünberg and Boike, 2019).

5) **Tussock** tundra is mostly located in flat, poorly drained areas in our study region. At our site specifically, 95% of the tussock patches have a slope of less than 4°, which is by far the lowest value of all vegetation types. Tussock–forming sedges such as cotton grasses (*Eriophorum*) and *Carex* species dominate. However, a variety of dwarf shrubs such as dwarf birch, willows, Labrador tea, mountain cranberry, bilberry (*Vaccinium uliginosum*), crowberry (*Empetrum nigrum*), bearberry (*Arctostaphylos uva-ursi*), and cloudberry (*Rubus chamaemorus*) are also present and mosses can be found between the tussocks. The vegetation height is 10–30 cm in general. Tussock tundra (class G4 (tussock-sedge, dwarf-shrub, moss

tundra) in Walker et al. (2005)) is the most abundant vegetation type in the study area with roughly 37% coverage (Grünberg and Boike, 2019).

**6) Lichen** tundra is dominated by lichen and dwarf shrubs 3–15 cm high, including mostly Labrador tea, mountain cranberry, crowberry, bearberry, and cloudberry. Some graminoids may also be present. Lichen tundra (class S1 (erect dwarf-shrub tundra) in Walker et al. (2005)) covers about 10% of the study area (Grünberg and Boike, 2019).

The approximate current spatial distribution of the six vegetation types and water is shown in Figure 2. The extent of tall shrub cover has expanded in the past and is likely to increase in the future at the Trail Valley Creek study site (MacKay, 1995; Lantz
et al., 2010).

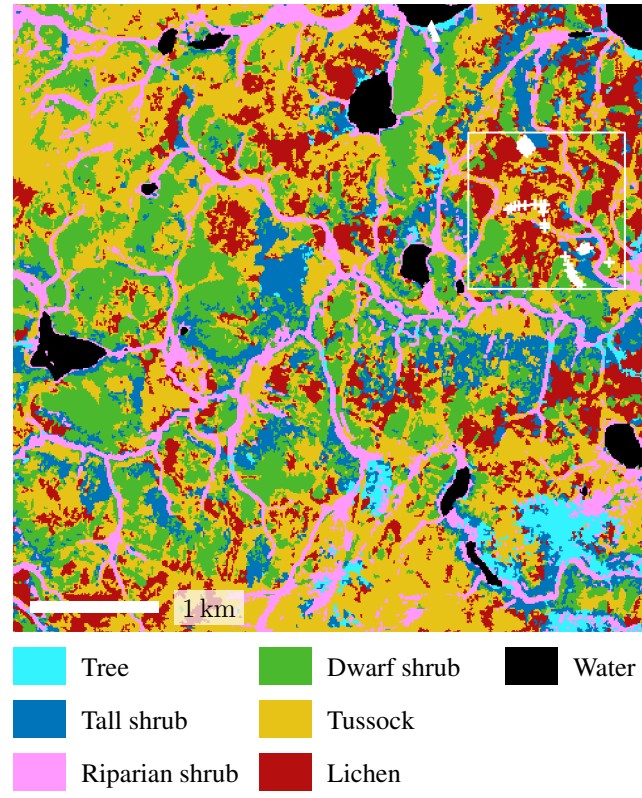

**Figure 2.** Vegetation map of the study region using the vegetation types as described above (Grünberg and Boike, 2019); the map is based on airborne orthophotos and vegetation height derived from airborne laser scanning (Anders et al., 2018); the white square indicates the study site extent including all sensor locations (white plus signs) as shown in Figure 1b.

## 2.2 Field measurements

Topsoil temperature was measured for two years at 68 locations distributed below different vegetation types at a maximum distance of 1200 m between the sensors in the study area (Figure 1b). We used iButton® temperature loggers (DS1922L) at

11 bit ($0.0625\,°C$) resolution with an accuracy of $0.5\,°C$ provided by the manufacturer. The sensors are $17\,mm$ in diameter and $6\,mm$ thick. Before deployment, we coated them with plastic to ensure water resistance. We tested all sensors in an ice bath and verified the temperatures measured at the zero curtain. As all sensors had zero-curtain temperatures between $-0.12\,°C$ and $0.30\,°C$, we assume that the accuracy provided by the manufacturer is realistic.

We installed all temperature loggers as close as possible to the surface (soil, moss, or lichen) but deep enough to be protected from solar radiation (typically $1$–$3\,cm$ deep). The measurement interval was every three hours; the measurement periods were 27 August 2016, 17:00 (local time) to 03 September 2017, 08:00 and 04 September 2017, 05:00 to 22 August 2018, 14:00. In between the two measurement periods, the sensors had to be removed in order to obtain data records and were reinstalled at the same positions. We peeled off the coating, read out the data, and used tape as the new coating, as the plastic could not be reapplied in the field setting. This was done in the field to assure that each sensor was reinstalled as close as possible to the original location. However, the coating was different and possibly the micro-location and the contact to the surrounding moss, lichen or soil also differed slightly. Thus, we treated the two periods separately in our statistical analysis.

We classified the vegetation type into one of the six categories described above at each topsoil temperature location based on observations in the field and pictures taken while installing and removing the sensors. Maximum vegetation height was measured from 04–06 August 2019 within $30\,cm$ around the sensor position.

Depth to the frozen soil was measured on 22 and 23 August 2018 three times around each topsoil temperature location within a $30\,cm$ radius. We averaged the three values before continuing with the statistical analysis. Thawing of the soil potentially continues until freeze–back starts in October; thus, maximum active layer thicknesses were not recorded. Snow depth was measured on 30 April 2017 approximately at the topsoil temperature locations based on a hand-held GPS because the flags marking each location were buried below the snow. In 2018, snow depth was measured by calculating the difference between a digital elevation model (DEM) of the snow and a DEM of the bare ground. The snow DEM was created using structure from motion photogrammetry in the software Pix4Dmapper (Pix4D SA, 2019), using images taken by an eBee Plus RTK at $3.5\,cm/pixel$ resolution on 22 April 2018 (Mann, 2018). The $1\,m$ resolution bare ground DEM was created using airborne LiDAR acquired in 2008 (Hopkinson et al., 2009). The resulting $1\,m$ resolution snow depth raster was calibrated using 1370 Magnaprobe (Sturm and Holmgren, 2018) snow depth measurements taken between 19 and 25 April 2018.

## 2.3 Data analysis

We analysed the topsoil temperature data separately for the four seasons. The definition of seasons used in our statistical analysis is based on mean daily air temperatures of the last 20 years (1999–2018) at Trail Valley Creek (Environment and Climate Change Canada, 2019). To obtain a complete 20-year record, we gapfilled the Trail Valley Creek data using a piecewise linear regression with data from the climate stations in Inuvik, $45\,km$ further south. Furthermore, we smoothed the average annual cycle using a 7-day moving window (Figure A2). We defined winter as all days with average air temperature below $-15\,°C$ (06 November - 10 April) while summer was defined as air temperature above $8\,°C$ on average (10 June - 25 August). The periods in-between were defined as spring and autumn (Figure A2).

**Table 1.** Number of data series per vegetation type in each measurement period; period 1: 27 August 2016, 17:00 (local time) to 03 September 2017, 08:00 and period 2: 04 September 2017, 05:00 to 22 August 2018, 14:00.

| Vegetation type | Number of data series | |
| --- | --- | --- |
| | Period 1 | Period 2 |
| Tree | 3 | 3 |
| Tall shrub | 17 | 15 |
| Riparian shrub | 2 | 2 |
| Dwarf shrub | 15 | 17 |
| Tussock | 14 | 14 |
| Lichen | 10 | 9 |
| Total | 61 | 60 |

Before analysis, we checked the quality of all time series data during the two measurement periods and removed the series if (a) the data record had more than 10% data gaps (4 series), (b) the average summer topsoil temperature was more than $5\,°\mathrm{C}$ colder than air temperature indicating that the sensor was either buried too deep and affected by the permafrost or affected by running water (3 series), or (c) more than 5% of the single summer measurements were more than $7\,°\mathrm{C}$ above air temperature indicating additional sensor warming by direct solar radiation (8 series) (Figure A1). We used air temperature measurements by Environment and Climate Change Canada (2019) at Trail Valley Creek for this comparison. The number of remaining data series in each measurement period per vegetation type is listed in Table 1.

We calculated a variety of different characteristics for each topsoil temperature series, namely (I) the mean value for the whole year, each season, and each month, (II) the range of all three hourly values between the 10th and the 90th percentile for the whole year, each season, and each month, (III) the slope of a linear regression of all daily average values within each season, i.e. the rate of warming or cooling, (IV) the cumulative sum of positive degree days from the beginning of the melt season until the end of August, (V) the day of the year when the soil temperature first rose above $-0.5\,°\mathrm{C}$ which indicates the beginning of the thawing period, (VI) the day of the year when the soil temperature first rose above $0.5\,°\mathrm{C}$ which indicates the end of the thawing period, and (VII) the day of the year when the soil temperature first drops below $-0.5\,°\mathrm{C}$ which indicates the end of the freezing period in autumn. We used $-0.5\,°\mathrm{C}$ and $0.5\,°\mathrm{C}$ as thresholds instead of $0\,°\mathrm{C}$ to account for sensor uncertainty. We used Python 3.6 to analyse the topsoil temperature series. In boxplots, we always show the absolute minimum and maximum as whiskers, the 25th and 75th percentile as the box, and the median as a line within the box.

In terms of statistical analysis, we used linear models (lm in R) to calculate Pearson's correlation coefficients and linear regressions for all numerical variables and to determine whether the variables were significantly related at $\alpha = 0.05$. In a first step, we used data of all vegetation types. In the second step, we split the data into two subsets, one containing the upland vegetation types *lichen tundra* and *dwarf shrub tundra* and one containing all other vegetation types. To assess the relationship of vegetation type to topsoil temperature characteristics, active layer thickness, and snow depth, we used a linear model (lm

in R) to estimate the adjusted $R^2$ and the fraction of variance explained by vegetation type. Prior to the analysis, we excluded the vegetation types *tree* and *riparian shrub*, as we do not have enough measurement series for these types. We assessed the impact of vegetation type on topsoil temperature characteristics based on all data series from both measurement periods. By analysing the two periods separately in the statistical model we accounted for the systematic differences between the years. For example, the summer of 2017 was warmer and started earlier than the summer of 2018. When we were interested in the impact of vegetation type on active layer thickness and snow depth, we used all sensor locations, even those for which we removed the temperature record, for example due to missing data. We used R version 3.4 for all statistical analyses.

## 3  Results

We found complex spatial relationships between the different topsoil temperature characteristics, snow depth, active layer thickness, and vegetation. In general, the mean topsoil temperatures of the months between December and April were strongly correlated (Figures 3 and 4). This implies that locations with relatively cold December topsoil temperatures were generally still colder than average in April. Locations which were colder in winter corresponded to warmer topsoil temperatures in May; however, the correlation of May temperatures with winter temperatures was much stronger for the first period (September 2016 – August 2017) than for the second (September 2017 – August 2018) . The summer temperatures between June and September were also highly correlated. In the first period, November topsoil temperature correlated well with October values and thus belonged to autumn, while November correlated more strongly with the winter months in the second period. Due to the long winter, the mean annual topsoil temperature correlated strongly with the topsoil temperature of the winter months, while it was almost uncorrelated with the summer topsoil temperature. The end date of the spring thawing period was strongly related to winter temperatures, snow depth, and active layer thickness. Contrary to this, the end of thawing was not significantly related to summer topsoil temperature (Figures 3 and 4).

The annual mean topsoil temperature recorded by all single sensors varied between $-3.7\,°C$ and $-0.8\,°C$ in the first period and between $-3.7\,°C$ and $0.1\,°C$ in the second period. In both periods, we found both the warmest and the coldest values in tussock tundra. The substantial variability within single types contributed to the low fraction of variance explained by vegetation (Table 2). If all locations within each vegetation type were averaged, lichen tundra had the coldest topsoil temperature with $-2.6\,°C$ and $-2.3\,°C$ in the first and second period, respectively. The warmest average topsoil temperature of $-2.0\,°C$ and $-1.3\,°C$ in the first and second period were measured below tall shrubs. If we considered the approximate landscape fractions of each vegetation type (Section 2.1), we estimated a mean annual topsoil temperature of $-2.3\,°C$ and $-1.7\,°C$ in the first and second period, respectively.

### 3.1  Soil temperature and vegetation in autumn

The two years 2016 and 2017 had different meteorological conditions in autumn. While air temperature dropped gradually to $-7\,°C$ before the first snowfall in 2016 (Figure B1a), daily mean air temperature had only dropped to $0\,°C$ by the date of the

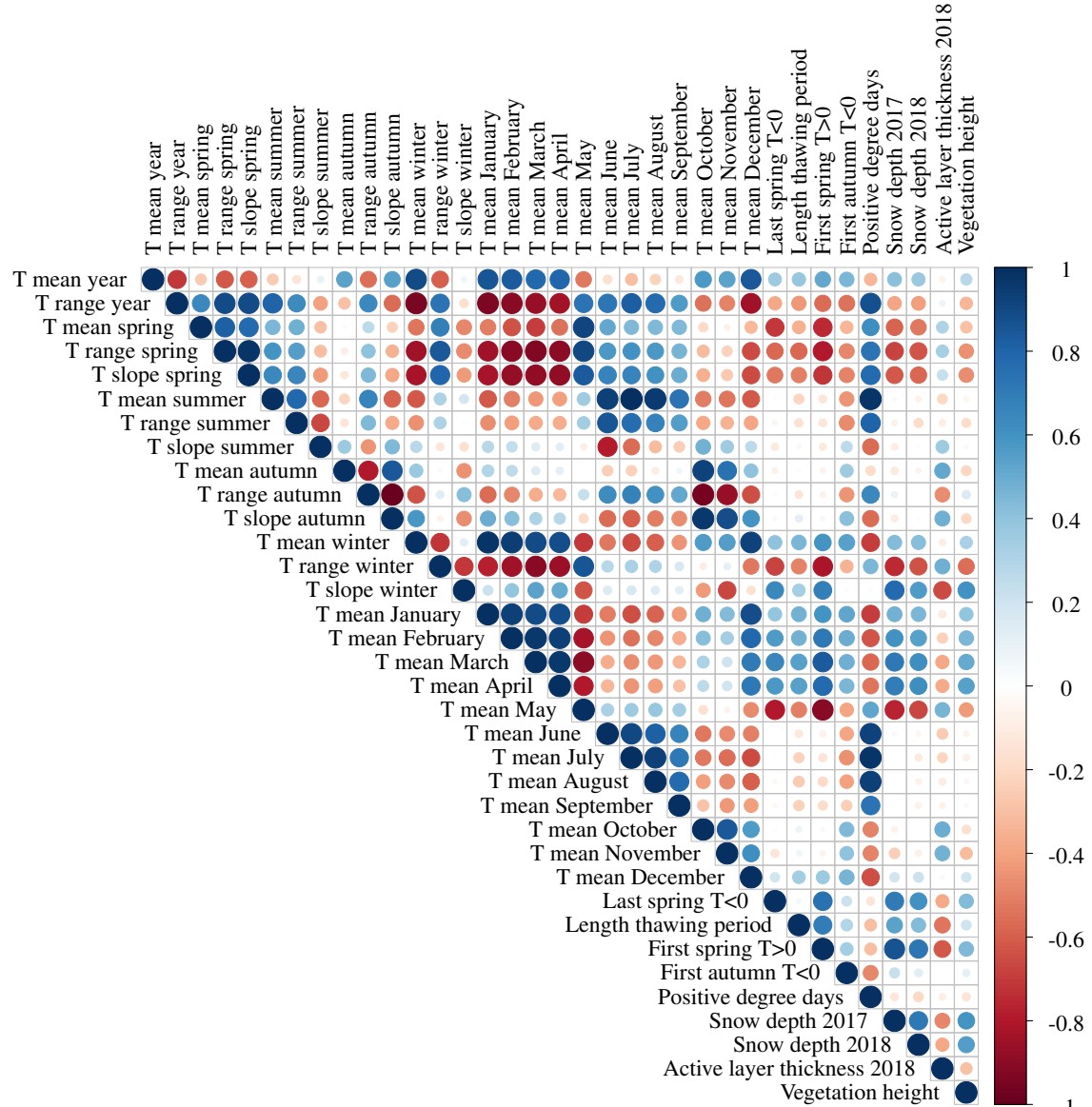

**Figure 3.** Pearson's correlation coefficients (R) of all topsoil temperature characteristics (°C), active layer thickness (cm), and snow depth (cm) in the first period (September 2016 to August 2017).

start of the snow accumulation in autumn 2017. Thus, the topsoil was significantly cooler in October 2016 as compared to 2017 (Figure 5d).

Autumn topsoil temperatures and cooling rates were not significantly related to vegetation type (Figure 5a–d, Table 2). The same was true for the start of the frozen period, which was almost identical at most sensor locations independent of vegetation type (Figure 5e). However, we observed considerable variability in the mean temperature within each vegetation type, at the

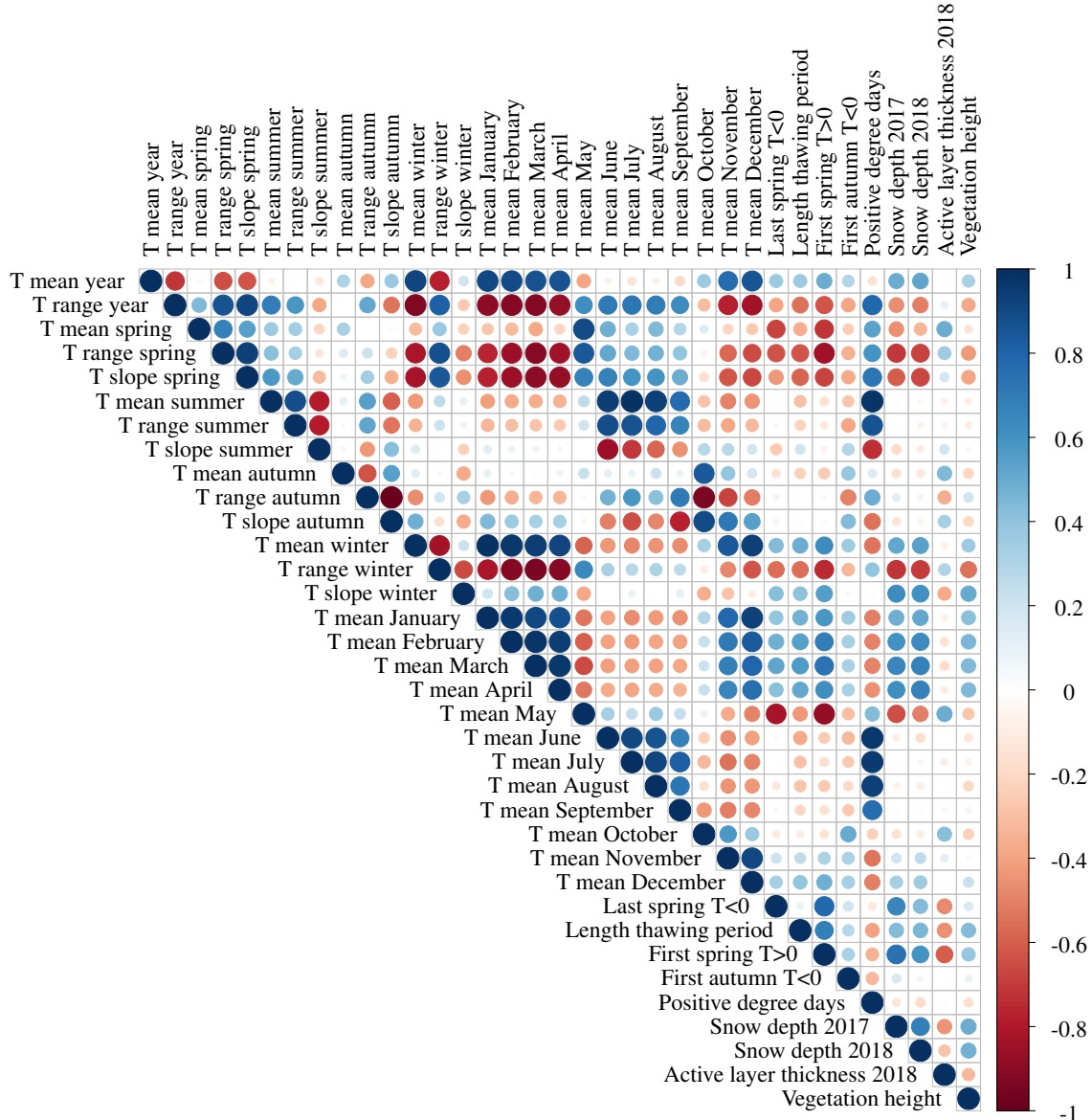

**Figure 4.** Pearson's correlation coefficients (R) of all topsoil temperature characteristics (°C), active layer thickness (cm), and snow depth (cm) in the second period (September 2017 to August 2018).

landscape level, and between the different years. Considering the complete autumn season (26 August – 05 November), the mean autumn temperatures varied between $-0.9\,°\text{C}$ and $1.2\,°\text{C}$ in 2016 and between $0.3\,°\text{C}$ and $1.7\,°\text{C}$ in 2017. For October only, the range of mean topsoil temperatures was even higher, between $-5.3\,°\text{C}$ and $-1.0\,°\text{C}$ for the coldest and warmest locations in the first period and between $-2.8\,°\text{C}$ and $0.2\,°\text{C}$ in the second period (Figure 5d). Similar differences could be

**Table 2.** Relationship of vegetation and topsoil temperature (T) characteristics, active layer thickness, snow depth, and vegetation height expressed by the fraction of variance explained by vegetation type in a statistical model (lm(response ~ period + vegetation type)) and the adjusted $R^2$; complete data set accounting for the measurement period of topsoil temperature first; results which are not significant at a level of 0.05 are shown in grey.

| Response | Fraction of variance | Adjusted $R^2$ |
|---|---|---|
| Mean annual T (°C) | 0.125 | 0.25 |
| Date of first T>0 °C | 0.548 | 0.55 |
| Cumulative degree days (°C) | 0.071 | 0.56 |
| Mean T May (°C) | 0.543 | 0.53 |
| Mean T July (°C) | 0.118 | 0.10 |
| Mean T October (°C) | 0.035 | 0.51 |
| Mean T March (°C) | 0.355 | 0.56 |
| T slope spring (°C/day) | 0.327 | 0.40 |
| T slope summer (°C/day) | 0.058 | 0.51 |
| T slope autumn (°C/day) | 0.053 | 0.09 |
| T slope winter (°C/day) | 0.432 | 0.56 |
| Active layer thickness 2018 (cm) | 0.342 | 0.31 |
| Snow depth 2017 (cm) | 0.699 | 0.68 |
| Snow depth 2018 (cm) | 0.584 | 0.56 |
| Vegetation height (cm) | 0.427 | 0.40 |

observed in the cooling rates, which varied most strongly within tussock tundra and tall shrubs. Some locations within these two types cooled down more than twice as fast as other locations (Figure 5c).

### 3.2 Soil temperature, snow, and vegetation in winter

There were significant differences in topsoil temperature between the two winter seasons (06 November - 10 April). Average topsoil temperatures were roughly 2 °C colder beneath all vegetation types in the winter 2016/2017 as compared to the following year. This observation agrees well with the difference in mean winter air temperature of $-20.3$ °C and $-18.2$ °C in the first and second period, respectively (Figure B1a). The extremely cold December of 2016 (6.5 °C colder than in the second period) led to a much faster soil cooling in the first winter as compared to the second (Figure B1a).

There was a strong relationship between vegetation type and topsoil temperature in winter. Topsoil under tall shrub tundra stayed warmest, followed by topsoil beneath tussock, dwarf shrub, and lichen tundra (Figure 6). Vegetation type explained 36% of the topsoil temperature variations in the coldest month (March) in a single year and 43% of the cooling rate during

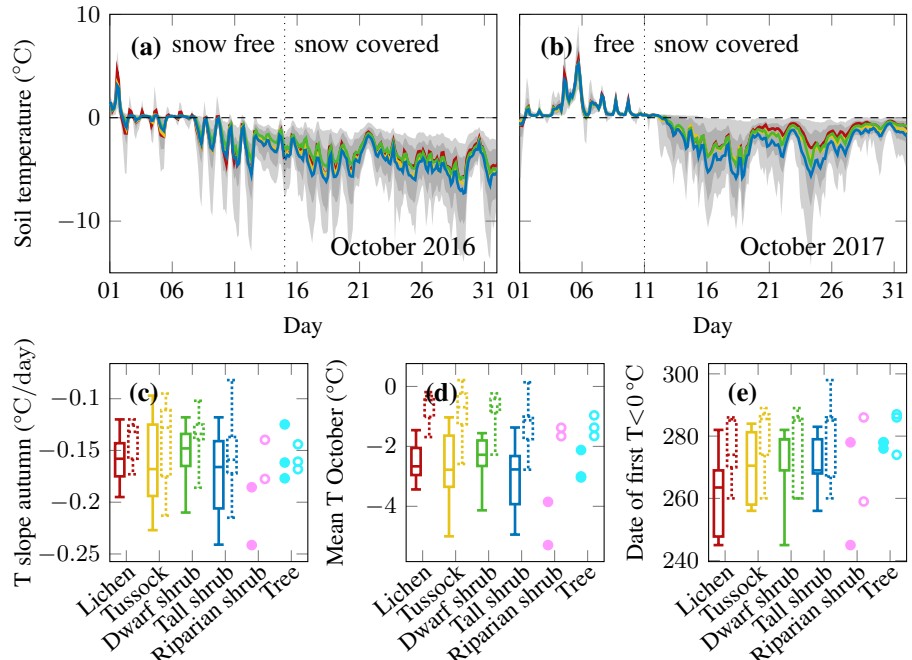

**Figure 5.** Topsoil temperature series of (a) October 2016 and (b) October 2017 representing autumn of the two different measurement periods; mean of all measurements below four different vegetation types in colour, the range of all single sensors in light grey and the range of all sensors between the 10th and 90th percentile in darker grey; the beginning of snow accumulation was derived from the Trail Valley Creek weather station albedo data; boxplots of all measurements per vegetation type with boxes of the first period left and the second period right and dotted; for riparian shrubs and trees all single observations are shown with filled symbols for the first period and open symbols for the second; (c) rate of cooling in autumn; (d) mean October topsoil temperature; (e) date when the freezing is completed.

winter (Table 2). The coldest mean temperatures were associated with the highest temperature variations below lichen tundra (Figure 6).

Snow depth strongly mediated the association between vegetation type and winter soil temperatures. Different vegetation types showed characteristic snow depth values (Figure 6e). The deepest snow cover was associated with tall shrubs, followed by tussock tundra and dwarf shrubs while lichen tundra was characterised by the shallowest snow cover. Vegetation explained 70% and 58% of the observed snow depth variability in 2017 and 2018, respectively (Table 2). The predictability of snow depth from vegetation type is limited by the differences in snow cover in different years. For example, the snow depth at the end of April 2017 reached higher maximum and lower minimum values as compared to 2018 indicating a stronger snow redistribution. The correlation between snow depth values of the different years was $R^2 = 0.49$ (Figure B2).

Several winter and spring topsoil temperature characteristics were also very strongly related to snow cover, in particular the day of the year when the topsoil warmed above $0\,°C$, the topsoil temperature slope during winter cooling, and the mean

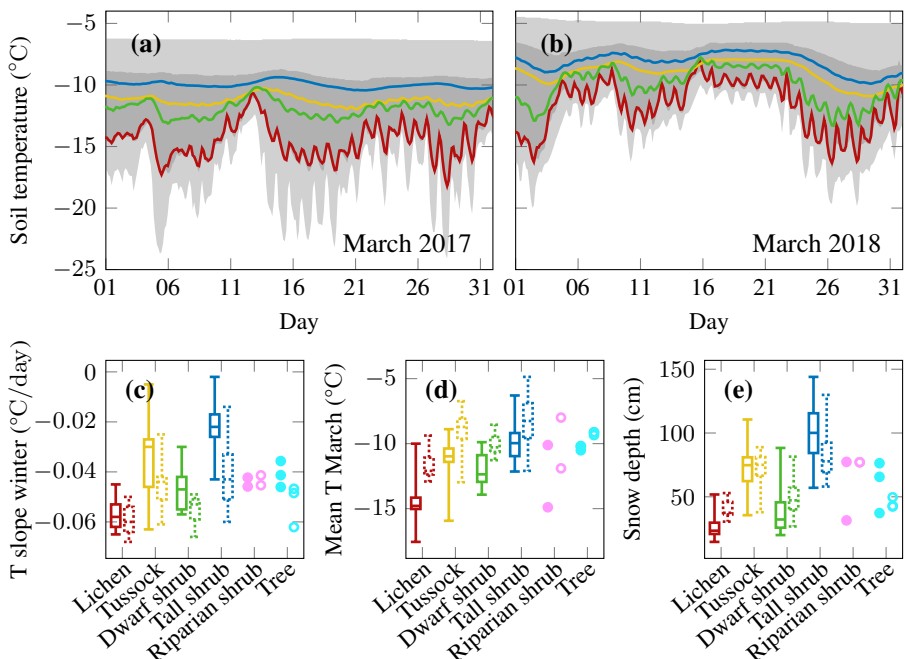

**Figure 6.** Topsoil temperature series of (a) March 2017 and (b) March 2018 representing the coldest topsoil temperatures in winter of the two different measurement periods; the mean of all measurements below four different vegetation types is shown in colour, the range of all single sensors is shown in light grey, and the range of all sensors between the 10th and 90th percentile is shown in darker grey; boxplots of all measurements per vegetation type are shown with boxes of the first period on the left and boxes representing the second period on the right and dotted; all single observations for riparian shrubs and trees are shown by filled symbols for the first period and open symbols for the second; (c) rate of cooling in winter; (d) mean March temperature; (e) snow depth end of April.

temperature of the winter and spring months (Figure 7). In 2018, snow depth was less strongly related to vegetation or topsoil temperature characteristics than in 2017 (Figure 7, right column, Table 2).

## 3.3 Soil temperature, snow, and vegetation in spring

Starting from the beginning of May, the relationship between vegetation and topsoil temperature was opposite to that found
in winter (Figure 8a, b). In general, topsoil temperatures beneath all vegetation types were very similar in the last few days of April. The soil below lichen tundra warmed up first and most strongly and showed the most pronounced diel variation. Dwarf shrub tundra was still cooler in May and warmed up a bit more slowly than lichen tundra. Tussock tundra topsoil temperatures rose above $0\,°C$ even later; tall shrub temperatures were the last to reach positive temperatures in 2017 and were similar to tussock tundra in 2018. This order became apparent in the slope of spring temperatures, in the mean May temperatures, and
in the day of the year when the thawing period ended (Figure 8c–e). Using a statistical model, we found that vegetation type

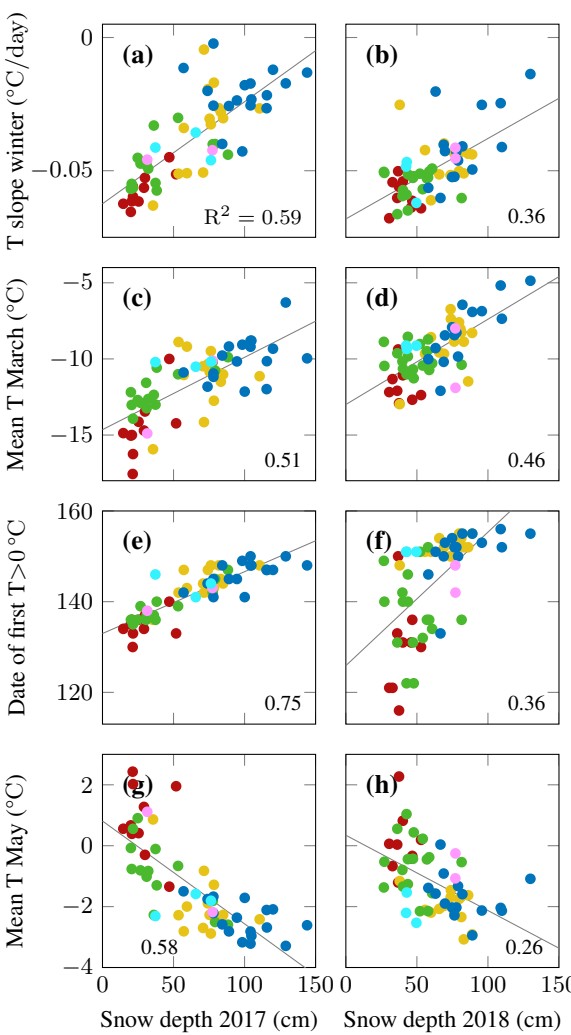

**Figure 7.** The relationship between snow depth end of April 2017 (left column) and 2018 (right column) and (a, b) rate of cooling in winter; (c, d) mean temperature in March; (e, f) day of year when the topsoil warms above $0\,°C$; (g, h) mean temperature in May; the symbol colours refer to different vegetation types (see e.g. Figure 6); the numbers indicate the Pearson's correlation coefficient $R^2$; the thin lines are regression lines of significant ($p < 0.05$) relationships.

explained 55% of the observed variability in the end of the thaw date and the mean May temperature and 33% of the variability in the spring warming rate (Table 2).

Furthermore, snowmelt timing was different between the years. Our observations of topsoil temperature indicated that the snowmelt period of the entire landscape was 20 and 40 days long in 2017 and 2018, respectively (Figure 8e). This observation was in contrast to the more spatially variable snow distribution observed in 2017 (Figure B2). The long melt period in 2018 was associated with a $3.4\,°C$ colder mean May air temperature and more cold spells as compared to May 2017 (Environment and

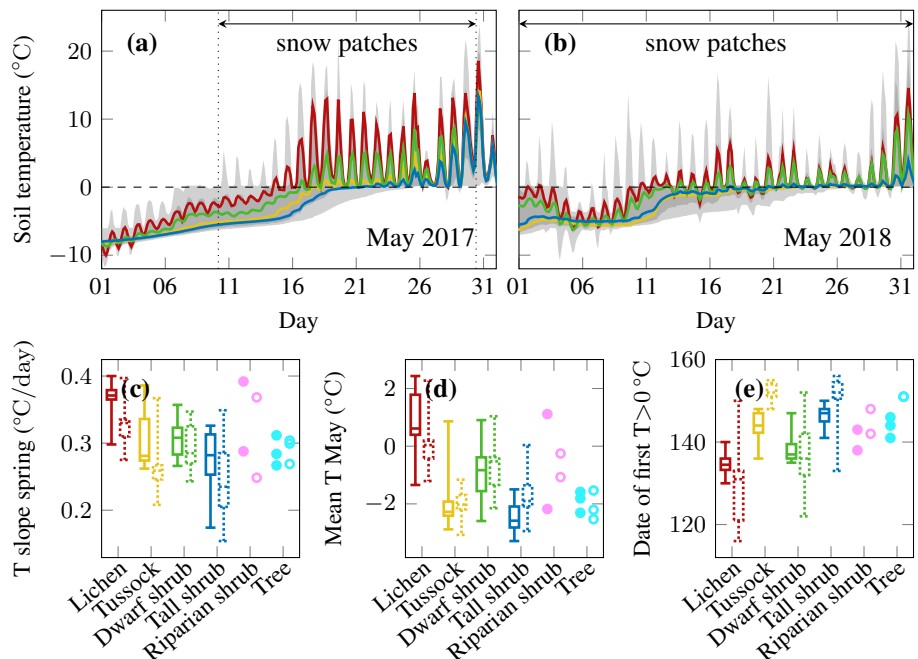

**Figure 8.** Topsoil temperature series of (a) May 2017 and (b) May 2018 representing spring of the two different measurement periods; mean of all measurements below four different vegetation types is shown in colour, the range of all single sensors is shown in light grey, and the range of all sensors between the 10th and 90th percentile is shown in darker grey; boxplots of all measurements per vegetation type are represented by boxes of the first period on the left and the second period on the right and dotted; for riparian shrubs and trees all single observations are shown with filled symbols for the first period and open symbols for the second; (c) rate of warming in spring; (d) mean May temperature; (e) date when thawing is complete.

Climate Change Canada, 2019) (Figure B1). However, due to the differences in snowmelt timing, the cold May air temperatures in 2018 did not necessarily translate into cold topsoil temperatures. While lichen tundra measurements indeed revealed colder May topsoil temperatures in 2018, both years were similar for dwarf shrub soil and May 2018 was associated with warmer topsoil temperatures below tussock tundra and tall shrubs.

## 3.4    Soil temperature, active layer thickness, and vegetation in summer

Topsoil temperatures in summer were more similar for all vegetation types than spring or winter temperatures (Figure 9a, b). In summer 2017, dwarf shrub tundra had the coolest summer temperatures. The other three vegetation types (excluding tree/riparian shrub) had almost identical mean values and the distributions largely overlapped (Figure 9c, d). In 2018, the difference between dwarf shrubs and the other vegetation types was much smaller and the mean summer slope was almost equal for lichen, dwarf shrub, and tall shrub tundra, while tussock tundra warmed at a slightly lower rate. Some locations had negative summer slopes in 2018 as the end of summer was relatively cool. The temperature difference between the two years

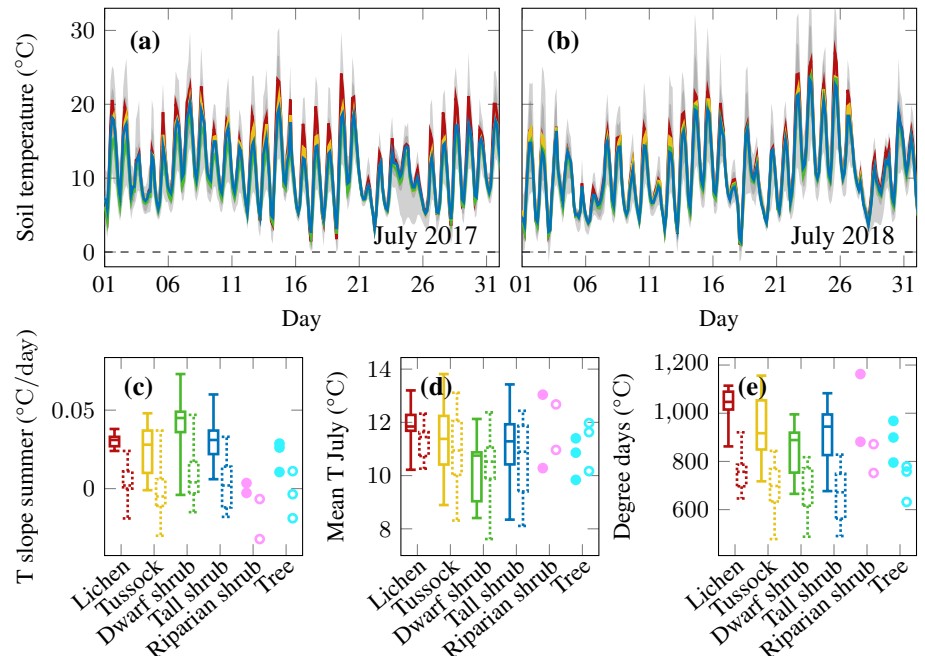

**Figure 9.** Topsoil temperature series of (a) July 2017 and (b) July 2018 representing summer of the two different measurement periods; mean of all measurements below the four different vegetation types is shown in colour, the range of all single sensors is shown in light grey, and the range of all sensors between the 10th and 90th percentile is shown in darker grey; boxplots show all measurements per vegetation type with boxes of the first period on the left and the second period on the right and dotted; for riparian shrubs and trees all single observations are shown with filled symbols for the first period and open symbols for the second; (c) rate of warming in summer; (d) mean July temperature; (e) cumulative sum of positive degree days until the end of August.

was much larger than the variability between vegetation types. The 2018 mean summer air temperature was $2.7\,°C$ cooler than in 2017. This difference translated into topsoil temperatures which were, on average per vegetation type, $1.3$–$2.2\,°C$ colder in 2018, with the smallest difference for dwarf shrubs. The small difference between vegetation types was also reflected in the low fraction of July mean temperature variance (12%) that could be explained by the vegetation type (Table 2). Lichen tundra featured slightly higher cumulative degree days than the other three vegetation types, which all showed similar values on average (Figure 9e). Even though the influence of vegetation type on cumulative degree days was statistically significant, vegetation only explained 7% of the observed variability (Table 2).

There was a weak association between active layer thickness and vegetation type. On average, we found deeper active layers below lichen and dwarf shrub tundra as compared to tussock and tall shrub tundra (Figure 10i). The variability in active layer thickness within each vegetation type was substantial. Correspondingly, vegetation was only weakly related to active layer thickness at the end of summer as it explained only 34% of the variability between different locations (Table 2).

The relationship between active layer thickness and topsoil temperature characteristics depended on vegetation type. If we considered all vegetation types jointly, active layer thickness was most strongly related to the date when the topsoil temperature

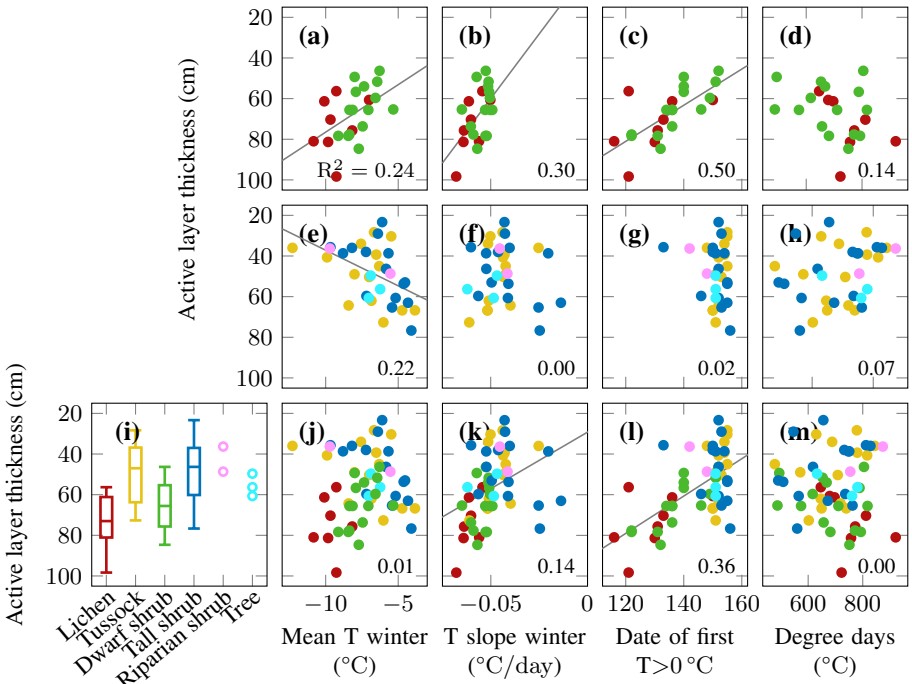

**Figure 10.** The relationship between active layer thickness at the end of August 2018 and (i) vegetation type; (a, e, j) mean topsoil winter temperature; (b, f, k) rate of topsoil cooling in winter; (c, g, l) day of the year when the topsoil warms above $0\,°C$; (d, h, m) cumulative sum of positive degree days until the end of August of the same year; top row: lichen and dwarf shrub tundra; middle row: tussock, tall shrub, riparian shrub, and tree; bottom row: all vegetation types; the numbers indicate the Pearson's correlation coefficient $R^2$; the thin grey lines are regression lines of significant ($p < 0.05$) relationships.

first rose above $0\,°C$ in spring (Figure 10l). While a moderate association existed with other spring, autumn, and winter characteristics, such as the cooling rate in winter (Figure 10k), summer characteristics did not show a significant correlation with active layer thickness when all vegetation types were considered (Figure 10m). If we split the data in two subsets, lichen and dwarf shrub tundra revealed very different winter characteristics as compared to the other vegetation types. While cold
winter temperatures were associated with deeper active layers in the following summer for lichen and dwarf shrub tundra (Figure 10a), tall shrub and tussock locations with cold winter temperatures developed shallow active layers (Figure 10e). The spring temperature characteristics were not significantly related to active layer thickness for tall shrub and tussock locations (Figure 10g). Conversely, the date of the first $T>0\,°C$ also apparently explained a moderate fraction of the active layer thickness variance within lichen and dwarf shrub tundra (Figure 10c). Higher positive degree days were associated with deeper active
layers below dwarf shrub and lichen tundra (Figure 10d), however, this relationship was not statistically significant (p=0.08). We observed significantly higher topsoil temperatures in October at locations with deep active layers (Figures 3, 4).

## 4   Discussion

In this study, we analysed how local variability of vegetation cover relates to topsoil temperature, active layer thickness, and snow depth at one site at the Arctic forest–tundra transition. We found distinct vegetation effects in the four seasons, which were partly counteracting each other in terms of the overall topsoil temperature and active layer thickness variability. In particular, low vegetation lead to an inverse relationship between winter temperature and active layer thickness and summer topsoil temperature did not influence active layer thickness significantly.

In agreement with Kropp et al. (2020) we found that winter topsoil temperatures controlled annual mean temperatures, and that topsoil below tussocks and tall shrub tundra was generally warmer than topsoil below short-statured vegetation. Furthermore, we found that topsoil temperature was most variable in space under the snow cover in winter and during snowmelt in May, while the spatial variability was less pronounced in summer and autumn, which agrees well with Gisnås et al. (2014).

### 4.1   Vegetation and soil temperature in relation to autumn processes

Vegetation type did not play a significant part in the autumn topsoil temperature variation (Figure 5), which agrees well with Romanovsky and Osterkamp (1995) who observed very similar dates of the start of ground freezing across three sites up to 63 km apart. On the other hand, Frost et al. (2018) found that tall shrubs experienced a delayed start of freezing by almost one month as compared to other vegetation types. We found that at all locations the topsoil temperature signal was dominated by air temperature and snowfall timing. The importance of snowcover onset for soil temperature was highlighted by many authors, including Ling and Zhang (2003); Zhang (2005).

While all vegetation types had similar average autumn temperatures and cooling rates, we observed considerable variability within single types as well as between years. The temporal variability of the start of freezing was described by Romanovsky and Osterkamp (1995), who found up to 18 days difference in the start of freezing date at a single location between 1987 and 1992. In their study, the temporal variability was higher than the spatial variability although the stations were up to 63 km apart. On the other hand, we observed considerable local variability of up to 39 days difference in the timing of the first below zero topsoil temperatures in a single year (Figure 5e). This variability is potentially driven by the soil cover of mosses or lichen, soil moisture, and microtopography rather than vascular plants. The importance of soil moisture in the freezing period was highlighted by Morse et al. (2016). We observed the largest range of mean October topsoil temperatures in tussock tundra (Figure 5d) which may be caused by a large range of moisture conditions depending on the microtopography in this vegetation type. However, it should be noted that variations in the sensor depth will play a major role in autumn when steep temperature gradients can be expected.

### 4.2   Vegetation and soil temperature in relation to winter snow processes

In winter, warm soil temperatures can be expected at locations with more snow such as below tall shrubs and in poorly drained and somewhat sheltered tussock-dominated areas (Lantz et al., 2009; Frost et al., 2018). The snow observed at our measurement locations was indeed deepest in tall-shrub areas, where soils were also the warmest in winter (Figure 6d, e). Across all

vegetation types, the differences in snow depth influenced the winter topsoil temperatures, temperature variability, and cooling
rates. Winter temperatures were coldest and most variable at vegetation types associated with low snow, in particular below
lichen tundra and, to a lesser extent, below dwarf shrubs (Figure 6a, b, d).

The variability of snow depth within vegetation types (Figure 6e) is likely due to (I) topography and microtopography
influencing the deposition of blowing snow (Essery and Pomeroy, 2004; Morse et al., 2012), (II) vegetation height and density
differences within one vegetation type (Sturm et al., 2001a; Essery and Pomeroy, 2004; Sturm et al., 2005a) and (III) factors
other than snow limiting shrub growth such as poor soil, too much or too little soil moisture, disturbances, and slow colonisation
(Swanson, 2015).

We also observed variability in the relationship between snow depth and winter topsoil temperature characteristics such as
the soil cooling rate and the mean and range of temperature values in the winter months (Figure 7a–d). The topography of
the site is gentle and cannot account for major differences in cold-season insolation; therefore, the variability is likely largely
due to differences in autumn soil moisture, snow density, and snow texture. Differences in snow density and texture across
the landscape can be caused by snow compaction at wind exposed sites, by loose snow accumulating within shrub canopies,
and by the formation of depth hoar. We do not have detailed snow observations at our measurement locations to analyse such
differences in detail. However, our results suggest that, for the same snow depth, topsoil temperatures in March are colder at
lichen sites as compared to dwarf shrub sites (Figure 7c, d). This agrees very well with the observation that lichen tundra can
be found at the most wind-exposed ridges. While historical surveys of snow density did not find differences between lichen
and dwarf shrub tundra at the site ($0.22\,\mathrm{g\,cm^{-3}}$, Wilcox et al. (2019)), snow texture and depth hoar formation likely differed
between shrub and shrub-free areas (Belke-Brea et al., 2020), leading to differences in the heat conductivity of each snowpack.

### 4.3  Vegetation and soil temperature in relation to spring processes

In spring, a thick snow cover delays soil warming and thus results in a reversal of the topsoil temperature – vegetation type
relationship. We observed similar topsoil temperatures beneath all vegetation types in the last few days of April. The soil below
tall shrubs was coldest in May, followed by the temperatures below tussocks and dwarf shrubs, while lichen tundra topsoil was
already the warmest (Figure 8d).

We found that the date when the topsoil warmed above $0\,^{\circ}\mathrm{C}$ was strongly related to vegetation type in 2017, and less so in
2018 (Figure 8e). A strong relationship can be expected due to the influence of vegetation on snow depth, snow density, and
snowmelt energetics (Pomeroy et al., 2006; Wilcox et al., 2019). The weaker relationship in 2018 is likely due to the more
complex spring weather patterns including multiple periods of warming and subsequent freezing periods. The colder May air
temperature in 2018 was associated with colder topsoil temperatures below lichen tundra and warmer topsoil temperatures
below tall shrubs, where the snow melted later as compared to May 2017 (Figure 8d). This indicates a first-order control of
snow depth as a buffer between air and topsoil temperature.
In general the presence of shrubs enhances snowmelt as soon as branches stick out of the snow, thus reducing surface
albedo and increasing long-wave emissions (Pomeroy et al., 2006; Marsh et al., 2010). Wilcox et al. (2019) made extensive
measurements of snow-free date with a drone at the same study site in 2016 and showed that dwarf shrub areas become snow

free earlier than non-shrub areas, regardless of snow depth and hillslope aspect. However, we did not observe this relationship at our locations, likely due to the small number of points we measured as compared to Wilcox et al. (2019). Instead, we found that topsoil warmed above $0\,°C$ slightly earlier at the lichen tundra locations, even though they were similar in snow depth to the dwarf shrub locations (Figure 7e, f).

## 4.4   Vegetation and soil temperature in relation to summer processes and active layer thickness

Although shrubs may reduce summer soil warming through shading and evapotranspiration (Pearson et al., 2013; Frost et al., 2018), in our study, the summer difference between all vegetation types is very small, and shrub tundra is comparable to tussock tundra in terms of topsoil temperature (Figure 9 and B3). Lichen tundra topsoil warms up slightly more, in particular during mid-day, but the average difference is less than $1\,°C$. These generally small differences in topsoil temperature likely contribute to the weak relation between vegetation type and summer temperature characteristics (Figure 10i, m, Table 2).

On the other hand, winter and spring temperature characteristics such as the date when the topsoil warmed above $0\,°C$, are strongly related to active layer thickness (Figure 10). At first glance, our data suggest that the snow-free date and thus the length of the summer period is the most important driver for active layer thickness (Figure 10l). The strong influence of snow-free date on active layer thickness has been highlighted in several other studies (Chapin et al., 2005; Wilcox et al., 2019). However, if we consider single vegetation types, the importance of snowmelt timing is strongly reduced. In particular, for tall vegetation (tall shrubs, riparian shrubs, trees) and tussock tundra we did not observe any correlation between snow-free date and active layer thickness at the end of summer (Figure 10g).

While mean winter temperature is not significantly related to active layer thickness at the landscape scale, we found opposite effects for upland vegetation (lichen and dwarf shrubs) as compared to tall vegetation and tussocks (Figure 10a, e, j). Warm winter temperature is unexpectedly associated with a shallow active layer in upland areas, while warm winter temperature is related to deep active layers below tall vegetation and tussocks (Figure 10a, e). This agrees well with results by Morse et al. (2012) from the outer Mackenzie Delta. They found that deep snow was associated with thick active layers in alluvial sites whereas it was associated with thin active layers in upland terrain. The opposite response of active layer thickness to mean winter temperature (and other winter and spring characteristics) below different vegetation types may be an artefact of the spatial correlation between snow depth and soil properties (Loranty et al., 2018). For instance, both snow depth and organic layer thickness tend to be shallow on exposed lichen-covered ridges, and the latter favours deeper active layers. As organic and moss layer thickness are important controls of active layer thickness (Fisher et al., 2016), the positive relationship between organic layer thickness and snow depth masks the actual effects of snow on the active layer thickness. Similarly, correlations between soil moisture, ice content, and snow depth may be different across the landscape, with variable effects on active layer development (Guan et al., 2010). It should also be noted that it was not recorded whether active layer thickness measurements were taken in a hummock or inter-hummock zone, which due to their vastly different soil properties, have a stronger effect on active layer thickness than any other variable (Wilcox et al., 2019). This reduces our ability to draw inferences from our active layer thickness measurements.

Summer topsoil temperature is only weakly related with vegetation type, which agrees well with Paradis et al. (2016). Vegetation type explains 12% of the variability of the summer topsoil temperature and 34% of the variability of active layer thickness in our study (Table 2), likely through its effects on snow depth. The generally weak relationships hinder upscaling approaches based on vegetation such as in Nelson et al. (1997); Widhalm et al. (2017). We found that thawing degree days are not correlated with active layer thickness (Figure 10m) at the landscape scale, echoing previous findings by Nelson et al. (1997). This is likely related to variable soil thermal properties and soil moisture. Within dwarf shrub and lichen tundra, where more uniform soil conditions may be expected, larger cumulative positive degree days are indeed associated with deeper active layers. In summary, our observations suggest that vegetation type is a better predictor of the near-surface thermal regime in winter than in summer in the Low Arctic. Furthermore, the soil temperature – active layer thickness relationship differs between dwarf shrubs and tall shrubs. Both vegetation types are currently expanding in the Arctic (Ropars and Boudreau, 2012; Tape et al., 2012) and our results indicate that their future distribution will govern the importance of summer versus winter processes for active layer thickness.

## 5 Conclusions

Based on topsoil temperature data from 68 sensors at a Low Arctic tundra site, we found large small-scale variability within and between vegetation types as well as between years and seasons. The spatial variation of the mean annual soil temperature was dominated by the winter signal. Autumn topsoil temperatures were dominated by atmospheric forcing and only weakly related to vegetation type. Conversely, vegetation type explained approximately one half of the variability in winter and spring soil temperature. An even stronger relation was observed between vegetation type and end-of-winter snow depth. Snow depth and, most likely, snow structural differences in space lead to pronounced differences in topsoil mean temperature and temperature variability in the winter and spring months and in snowmelt timing, all of which were strongly related to vegetation type. At the landscape scale, we found that active layer thickness was most strongly related to snowmelt timing. However, if we considered only specific vegetation types with presumably more similar soil conditions, mean winter temperature played a more important part. Unexpectedly, warm winter topsoil temperature was associated with shallow active layers below lichen and dwarf shrub tundra, whereas we found the opposite beneath other vegetation types. Summer topsoil temperatures were similar below all vegetation types and not significantly related to active layer thickness at the end of August. To conclude, vegetation can, with limitations, be used as a proxy for snow depth variability at the local scale, but it is a poor proxy for summer and autumn topsoil temperature or active layer thickness.

*Data availability.* The topsoil temperature data used in this study is published in Pangaea (doi:10.1594/PANGAEA.918615).

## Appendix A: Methods

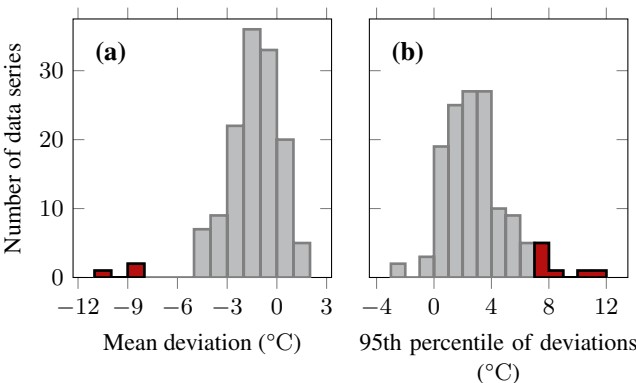

**Figure A1.** Histograms of the deviation of topsoil temperature minus air temperature in summer; (a) mean and (b) 95th percentile of the deviations for each sensor in each measurement period; data series coloured in red were removed from the analysis because (a) the average deviation was less than $-5\,^{\circ}\mathrm{C}$ indicating that the sensor was either buried too deep and affected by the permafrost or affected by running water (3 series) or (b) more than 5% of the single summer measurements were more than $7\,^{\circ}\mathrm{C}$ above air temperature indicating additional sensor warming by direct solar radiation (8 series); air temperature data by Environment and Climate Change Canada (2019).

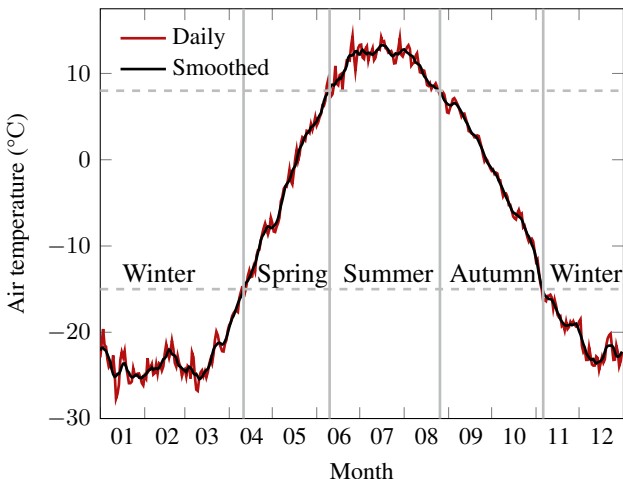

**Figure A2.** Mean annual cycle of air temperature at Trail Valley Creek for 1999–2018, gap-filled data series; daily values and values smoothed with a seven-day moving window as used for definition of the seasons; data by Environment and Climate Change Canada (2019).

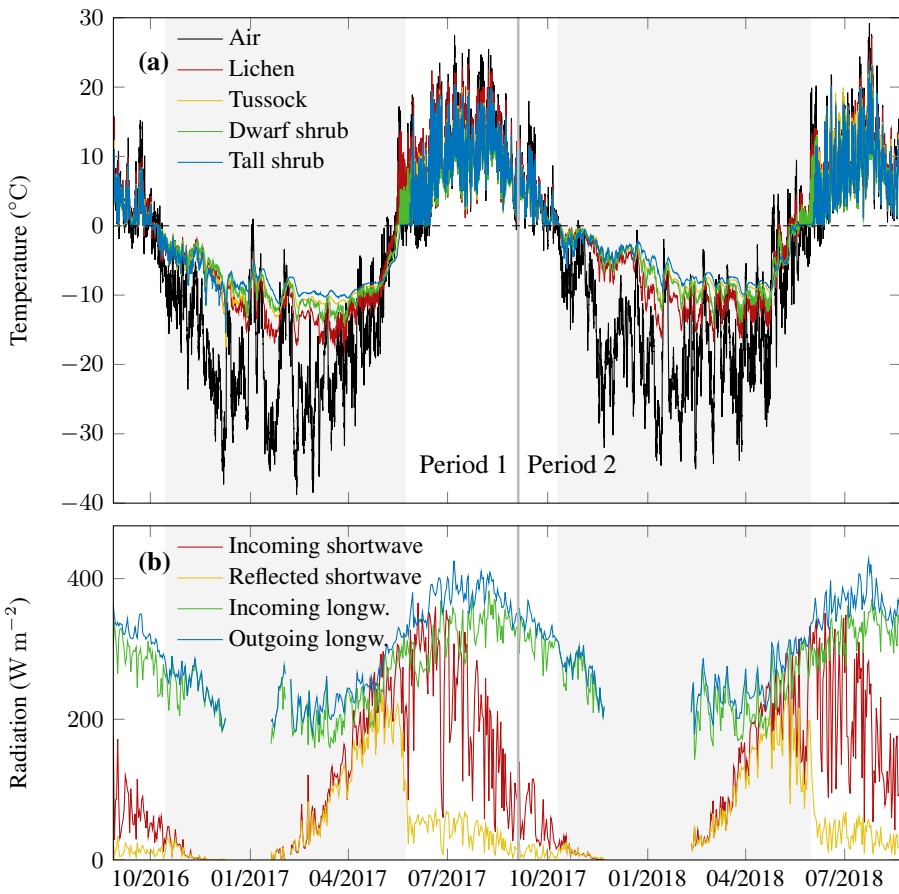

**Figure B1.** Meteorological conditions at the Trail Valley Creek weather station during the two study periods including snow cover estimated from daily albedo > 0.4 indicated as shaded area. (a) Air temperature by Environment and Climate Change Canada (2019) and median topsoil temperature of each vegetation type; (b) daily mean values of four component radiation.

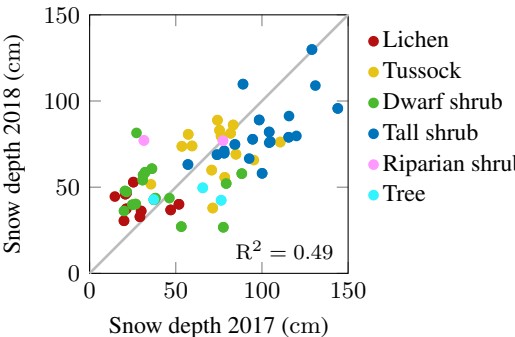

**Figure B2.** Snow depth measured at the end of April in 2017 and 2018 at approximately the same locations; the number indicates the Pearson's correlation coefficient $R^2$; the grey line is the 1:1 line.

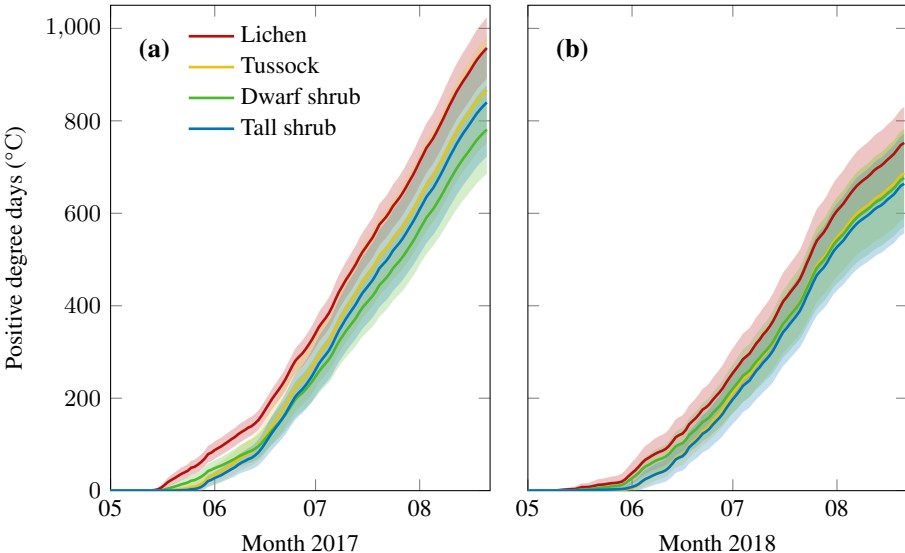

**Figure B3.** Cumulative positive degree days of topsoil temperature below different vegetation types; the lines indicate the mean of all series per type, the shaded areas represent the mean plus/minus the standard deviation; (a) 2017, (b) 2018.

*Author contributions.* J. Boike and I. Grünberg conceived the study. J. Boike, S. Zwieback, E. Wilcox, I. Grünberg, and several helpers carried out the field measurements. I. Grünberg analysed the data with input from J. Boike, E. Wilcox, and S. Zwieback. I. Grünberg prepared the manuscript with contributions from all co-authors.

*Competing interests.* We declare that no competing interests are present.

*Acknowledgements.* This contribution was financially supported by Geo.X, the Research Network for Geosciences in Berlin and Potsdam (Grant-number: SO 087 GeoX) and by funding from the Helmholtz Association in the framework of MOSES (Modular Observation Solutions for Earth Systems). We thank Cory Wallace, Branden Walker, Bill Cable, and Stephan Lange for helping with the data collection in the field.

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
