# Peer review of "Linking tundra vegetation, snow, soil temperature, and permafrost"

_Biogeosciences, 2020_

## Referee Comment (RC1) · Anonymous Referee #1 · 8 May 2020

The spatial distribution of near-surface temperature is characterized with respect to distinct Arctic tundra vegetation communities over the course of two years. Vegetation was found to influence snow cover throughout the autumn, winter, and spring with seasonally variable effects on soil temperature. However, soil temperature was not significantly correlated with vegetation type or active layer thickness during the summer.

General comments: It's well established that tundra vegetation affects snow depth/cover and vice versa with co-varying implications for soil moisture and temperature (see early mountain tundra work by Dwight Billings and Skip Walker among others). However, it's not as well known how these processes combine to affect the spatial variability of active layer depth in permafrost regions. The current study rigorously characterizes seasonal relationships between vegetation, snowpack, and soil temperature

in heterogeneous Arctic tundra, but fails to link these results to permafrost dynamics in a meaningful way. I believe the authors could make this connection by (1) re-framing the results and (2) elaborating on the broader impacts of this work as detailed below.

(1) It wasn't until I read the final discussion section that I internalized what I think is the main take home message and contribution of this work: small scale differences in vegetation and snow accumulation do not affect active layer thickness in this system. I think this (non-)result would generate considerable interest (and citations) if it were highlighted in the abstract, the first discussion paragraph, the conclusion, and potentially the title, but it's difficult to pull out this contribution with the current focus on vegetation-snow-temperature dynamics that are mostly known already. To put it another way, I think the rigorous soil temperature/veg/snow measurements could be leveraged to gain new information about active layer dynamics, as opposed to the main focus being on the measurements themselves.

(2) The broader impacts of the study are not well developed. Why does winter soil temperature matter if it has no bearing on active layer thickness during the summer? What are the implications for shrub expansion or other expected changes/disturbances in these systems? I found this information particularly wanting in the in the abstract and discussion sections. I see this as a natural follow on to (1) insofar as it's your opportunity to describe what the non-result means for future predictions of warming, permafrost degradation, greenhouse gas emissions, etc. This will increase the impact of your work by clearly demonstrating the scientific contribution to other scientists, the media, and the general public.

Specific comments: L15-16: "shrinking snow cover" means less snow-covered area or snow covers the same amount of area for less time?

L17-19: This is the type of broader impact statement that I was looking for in the abstract i.e., why does soil temperature matter?

L25-27: It would be helpful if you could describe the thermodynamic mechanism(s) for

this.

L35-36: And presumably to get additional moisture in some cases.

L40: Missing words here.

L60: Please explain how tall vegetation cools the soil through evapotranspiration. My first thought is that transpiration would dry the surrounding soil, which would lower its heat capacity and make it more vulnerable to summer warming.

L65: Wrong word "and".

Figure 1: What is the yellow area in (a)?

Figure 2: The colors used to denote the "tree" and "tall shrub" and the "riparian shrub" and "lichen" vegetation types are indistinguishable to me. I don't see the white plus on the map.

L139-140: I've re-read this statement several times now and still can't wrap my head around it. Please clarify with particular attention to double meanings associated with "mean deviation" and "less than" when referencing a negative number.

L173-174: Is this really the most accurate interpretation of Figure 3 and 4? It seems like the correlations are all over the place to me.

L184-185: Interesting.

L186: I thought the highest soil temps were 0.1C and -0.8C as stated on L183-184?

L206-207: I'd expect soil temp differences to be damped relative to air temp differences. How did snow cover compare between these sensors?

L214-215: I'm very curious as to why the lichen and dwarf shrub snow depth went up between 2017 and 2018 but the tall shrub and tussock snow depth went down during the same period. Okay now I see that you invoke wind redistribution on L222 – might be good to bring in here.

L218-219: Can you provide the regression stats to verify this?

L239-241: This speaks to the first-order control of snow depth as a buffer between air and soil temps.

Figure 7: The take away messages from this figure would be clearer if you showed significant trendlines in (b) through (e) and (g) through (j).

L270: I'm not sure how the first and second mentions of "site" on this line are related. Please clarify (and preferably expand), especially since the first two discussion sentences are prime real estate.

L274: Most variable between sensors or through time?

L278: Please elaborate on what's meant by "general pattern".

L279-280: This conjecture would be a lot stronger if it were supported with citations (there are many) as well as a process-based description, especially given the short length of this section.

L288-291: Citations are required for everything you wish to invoke here.

L312: And increasing long-wave emissions.

L319: Same comment as L60.

L319-322: So none of it matters for permafrost? I think you've buried the lede here. This non-result should be highlighted as a main conclusion of your work (see general comments).

L330-337: This argument is hard to follow.

---

## Referee Comment (RC2) · Anonymous Referee #2 · 18 May 2020

This paper uses two years of topsoil temperature, snow, and active layer data from six vegetation types within a heterogeneous Low Arctic landscape in the northwestern Canadian Arctic to evaluate the relationships among these variables – vegetation, snow, soil temperature, and active layer depth. With changes in arctic vegetation being readily observed, there needs to be a greater understanding of how vegetation influences snow dynamics, ground temperature, and ultimately active layer depth and permafrost. Several papers in the literature do exist on this topic, however, the results collectively are not incredibly clear and consistent, and more data and analyses are needed. With a soil temperature dataset as robust as the one from this study, there are unlimited ways to analyze the data, and everyone will have their own opinion on how best to do that. The analysis presented here is generally a fine one, and is informative, and therefore I will not give opinions on other ways in which the analysis could have been done, but rather will provide some constructive comments on the existing analysis. The following are some general, more specific, and some minor editorial suggestions: 1) State the depth of the data loggers in the Abstract 2) Line 28 – Low Arctic is commonly capitalized, and change the "an" to "a" 3) Line 37 - You mention that tussock tundra is commonly found in depressions, which is not always my experience, and I'm not sure how widespread that it. Certainly not on the exposed hilltops, but more often I think that tussock tundra is found on mesic slopes, as opposed to more saturated lowland positions. 4) Line 40 – should be "active layer thickness at the end of summer" 5) Line 88 – "gasses" should be "grasses" 6) Lines 139-140 – since you use "summer" here, I was wondering if you should define your seasons first. Also wondering if there is a better way of stating this – e.g. average topsoil temperature was more than 5 degrees C lower than air temperature. 7) Lines 140-142 – how were these values determined, i.e. the -5 and +7 degrees C differences? 8) Line 148 – Assuming that you smoothed the data prior to defining the seasons? Did smoothing remove all spikes in the data such that the winter and summer temperature thresholds yielded continuous seasons, i.e. were there any days within winter or summer that fell outside of the threshold temperatures? 9) Lines 153-154 – I understand how the slopes for fall and spring might work directionally as cooling and warming respectively, but I would imagine that the peak warm and cool days with fall somewhere in the middles of summer and winter respectively, and therefore the slopes would not be very informative for these seasons. 10) Lines 157-158 – Would the day when temperatures first drop below 0.5 degrees C be the beginning of the freezing period in autumn? 11) Line 168 – remove the comma after "those" 12) Line 181 – by "date of thawing" do you mean first date, end date, or both? 13) Line 199 – give the actual temperature ranges 14) Lines 263-264 – with regard to the relationship between October soil temperatures and active layer depth, I have a hard time believing that Oct. temperatures can influence active layer depth – more likely the other way around maybe? 15) Figure 8 – I'm not sure that this figure is very useful. If vegetation is influencing snow depth, than it's

very likely that snow depth in one year will be related to snow depth the next year. You might considering removing this. 16) Figure 7– you discuss these relationships a lot, but I don't see any statistical analyses on them (with the exception of panels (a) and (f)). Are these significant relationships across and within vegetation types? Also, if these relationships are considered to be causal, then snow depth should be on the x-axis, as it is driving the other variables (again, except panels (a) and (f)). 17) Figure 11 – same as Figure 7 with regard to the statistical analyses (except panel (a)). Axes are fine in this figure as active layer depth is the assumed dependent variable. 18) Line 327 – remove the word "temperature." 19) Lines 332-334 – examples of where relationships are being discussed with no statistical analyses 20) Finally, with regard to the Discussion and Conclusions, the vegetation may indeed have an effect on active layer depth, through its effect on snowpack, as vegetation is effecting the snowpack, which is driving snowmelt and spring/summer temperature regimes.

---

## Author Comment (AC1) · 6 Jul 2020

**Answer to the reviewers comments on**
**Linking tundra vegetation, snow, soil temperature, and permafrost**

Inge Grünberg, Evan J. Wilcox, Simon Zwieback, Philip Marsh, Julia Boike

**Anonymous Referee #1**

We thank the anonymous referee #1 for his/her helpful and knowledgeable comments on our manuscript. Please find our answers to all specific points below in black after the original comments (in grey). We also attached the track-changed manuscript with new text in blue and text of the old version in red and small. We updated all figures as the reviewer requested a change of colours and some more details such as regression lines. We removed the old graphics from the track-changed version to limit the file size. The line numbers refer to the track-change version.

The spatial distribution of near-surface temperature is characterized with respect to distinct Arctic tundra vegetation communities over the course of two years. Vegetation was found to influence snow cover throughout the autumn, winter, and spring with seasonally variable effects on soil temperature. However, soil temperature was not significantly correlated with vegetation type or active layer thickness during the summer.

General comments:

It's well established that tundra vegetation affects snow depth/cover and vice versa with co-varying implications for soil moisture and temperature (see early mountain tundra work by Dwight Billings and Skip Walker among others). However, it's not as well known how these processes combine to affect the spatial variability of active layer depth in permafrost regions. The current study rigorously characterizes seasonal relationships between vegetation, snowpack, and soil temperature in heterogeneous Arctic tundra, but fails to link these results to permafrost dynamics in a meaningful way. I believe the authors could make this connection by (1) re-framing the results and (2) elaborating on the broader impacts of this work as detailed below.

(1) It wasn't until I read the final discussion section that I internalized what I think is the main take home message and contribution of this work: small scale differences in vegetation and snow accumulation do not affect active layer thickness in this system. I think this (non-)result would generate considerable interest (and citations) if it were highlighted in the abstract, the first discussion paragraph, the conclusion, and potentially the title, but it's difficult to pull out this contribution with the current focus on vegetation-snow-temperature dynamics that are mostly known already. To put it another way, I think the rigorous soil temperature/veg/snow measurements could be leveraged to gain new information about active layer dynamics, as opposed to the main focus being on the measurements themselves.

We thank the reviewer for this comment because it helped us to narrow our focus and stress the most important results. We rewrote the abstract including a more general motivation of our study, more detailed results on the summer processes, and implications for permafrost studies. We added more information to Figure 10. We also changed the first discussion paragraph (l.

328–332) and changed the order of the sentences in the conclusion. We also added more discussion on this result in Section 4.4 to emphasize the roles of winter versus summer processes for active layer thickness.

30    (2) The broader impacts of the study are not well developed. Why does winter soil temperature matter if it has no bearing on active layer thickness during the summer? What are the implications for shrub expansion or other expected changes/disturbances in these systems? I found this information particularly wanting in the in the abstract and discussion sections. I see this as a natural follow on to (1) insofar as it's your opportunity to describe what the non-result means for future predictions of warming, permafrost degradation, greenhouse gas emissions, etc. This will increase the impact of your work by clearly demonstrating

35    the scientific contribution to other scientists, the media, and the general public.

We added sentences on the broader picture and the implications of our work to the abstract (l. 20–24) and at the end of the discussion (l. 436–447). We also change the text in multiple other locations to be more precise; these changes are described below with the specific comments.

Specific comments:

40    **L15-16** "shrinking snow cover" means less snow-covered area or snow covers the same amount of area for less time? Both have been observed, decrease in snow cover extent and shorter snow covered periods. We specified this in the text now (l. 27).

**L17-19** This is the type of broader impact statement that I was looking for in the abstract i.e., why does soil temperature matter? We agree that our abstract did not include the broader picture nor the implications of our work. We moved the

45    sentence to the abstract and rephrased most other sentences (l. 1–4, 11–24).

**L25-27** It would be helpful if you could describe the thermodynamic mechanism(s) for this. We reordered the paragraphs to explain these mechanisms right after this sentence. Shrub cover mainly influences the soil temperature through soil shading in summer and snow trapping in winter (l. 61–64).

**L35-36** And presumably to get additional moisture in some cases. We agree and added a comment on this effect to the

50    manuscript (l. 45).

**L40** Missing words here. Added (l. 51)

**L60** Please explain how tall vegetation cools the soil through evapotranspiration. My first thought is that transpiration would dry the surrounding soil, which would lower its heat capacity and make it more vulnerable to summer warming. We agree with the reviewer that the heat capacity will be reduced in a dry soil. However, the reduction in thermal conductivity has

55    the opposite effect. Additionally, evapotranspiration of tall vegetation may contribute to soil cooling as it is an energy sink. We added this explanation and a reference (Fisher et al., 2016) to the introduction (l. 66–68).

**L65** Wrong word "and". Corrected (l. 82)

**Figure 1** What is the yellow area in (a)? The orange/yellow area is the tundra extent; we clarified this in the caption now.

**Figure 2** The colors used to denote the "tree" and "tall shrub" and the "riparian shrub" and "lichen" vegetation types are indistinguishable to me. I don't see the white plus on the map. We changed the colour of 'riparian shrub' in all figures from purple to pink and increased the size of the plus signs.

**L139-140** I've re-read this statement several times now and still can't wrap my head around it. Please clarify with particular attention to double meanings associated with "mean deviation" and "less than" when referencing a negative number. We rephrased the sentence in a simpler way using '5 °C colder' instead of 'mean deviation ... less than −5 °C' (l. 168).

**L173-174** Is this really the most accurate interpretation of Figure 3 and 4? It seems like the correlations are all over the place to me. We clarified that sentence to be more specific. We meant December temperatures are highly correlated with January/February/March/April temperatures (l. 204–206).

**L184-185** Interesting. We agree that this is interesting. The graph below summarises the variability of mean annual temperature within all vegetation types. Tussock tundra is most variable. In general, vegetation type is not a strong driver of mean topsoil temperatures as the boxes widely overlap. Although there is a significant relationship, vegetation only explains 12.5% of the variability of mean annual temperature (Table 2). We added this extra information to the text (l. 218–219).

[Figure]

**L186** I thought the highest soil temps were 0.1C and -0.8C as stated on L183-184? The first numbers were on individual locations, the second numbers are averaged by vegetation type. We clarified this in the text (l. 219).

**L206-207** I'd expect soil temp differences to be damped relative to air temp differences. How did snow cover compare between these sensors? We agree, that it could be expected that the soil temperature signal is dampened as compared to the air temperature. However, the strongest difference of 6.5 °C in air temperature was observed in December, with presumably still less snow than later during the winter. The extremely cold December of 2016 lead to a much faster soil cooling in the first winter as compared to the second. A part of this inter-annual difference was not used up when January 2017 was, on average, 1.8 °C warmer than 2018. We added the information on the cold December temperatures to the manuscript (l. 250–251).

**L214-215** I'm very curious as to why the lichen and dwarf shrub snow depth went up between 2017 and 2018 but the tall shrub and tussock snow depth went down during the same period. Okay now I see that you invoke wind redistribution on L222 – might be good to bring in here. We changed the order of the paragraphs (l. 261–264).

**L218-219** Can you provide the regression stats to verify this? The statistics for vegetation are listed in Table 2, 2nd and 3rd row from the bottom. We added the reference to the table (l. 272). For the other variables, we added Pearson's correlation coefficient and regression lines to Figure 7 (and for consistency also to Figure 10 (previously 11) and B2 (previously 8)).

**L239-241** This speaks to the first-order control of snow depth as a buffer between air and soil temps. We agree and added the comment to section 4.3 of the Discussion (l. 390–391).

**Figure 7** The take away messages from this figure would be clearer if you showed significant trendlines in (b) through (e) and (g) through (j). We added the regression lines in Figures 7 and 10 (previously 11) (for consistency).

**L270** I'm not sure how the first and second mentions of "site" on this line are related. Please clarify (and preferably expand), especially since the first two discussion sentences are prime real estate. The second sentence (and usage of 'site') was meant to clarify the limitations of the first sentence. We agree with the reviewer, that a more concise and bold statement is better and therefore replaced the sentence (l. 328–332).

**L274** Most variable between sensors or through time? Between sensors; we specified this now (l. 335).

**L278** Please elaborate on what's meant by "general pattern". We were referring to the characteristics of the time series; we specified this now (l. 343).

**L279-280** This conjecture would be a lot stronger if it were supported with citations (there are many) as well as a process-based description, especially given the short length of this section. We completely changed the discussion of autumn processes (Section 4.1, l. 339–350) and added references, in particular to the work by Romanovsky and Osterkamp (1995).

**L288-291** Citations are required for everything you wish to invoke here. We added citations to Essery and Pomeroy (2004); Morse et al. (2012); Sturm et al. (2001); Essery and Pomeroy (2004); Sturm et al. (2005); Swanson (2015) to relate the statement to other studies (l. 364–367).

**L312** And increasing long-wave emissions. Yes, we agree and added this information (l. 339).

**L319** Same comment as L60. We added the information to L60 and an additional reference in the discussion (Pearson et al., 2013) (l. 400) and the introduction (Fisher et al., 2016) (l. 66–68).

**L319-322** So none of it matters for permafrost? I think you've buried the lede here. This non-result should be highlighted as a main conclusion of your work (see general comments). We now highlighted this finding in the conclusions (l. 460–462) and in the abstract (l. 17–24).

**L330-337** This argument is hard to follow We rewrote this section, added more details and references (Morse et al., 2012; Fisher et al., 2016; Guan et al., 2010; Paradis et al., 2016) to this paragraph to clarify our findings on the snow–active layer relationships (l. 406–412,425–428).

**115 References**

[revised manuscript text omitted]

---

## Author Comment (AC2) · 6 Jul 2020

**Answer to the reviewers comments on**
**Linking tundra vegetation, snow, soil temperature, and permafrost**

Inge Grünberg, Evan J. Wilcox, Simon Zwieback, Philip Marsh, Julia Boike

**Anonymous Referee #2**

We thank the anonymous referee #2 for his/her helpful and knowledgeable comments on our manuscript. Please find our answers to all specific points below in black after the original comments (in grey). We also attached the track-changed manuscript with new text in blue and text of the old version in red and small. We updated all figures as the reviewer #1 requested a change of colours. We also added some more details such as regression lines. We removed the old graphics from the track-changed version to limit the file size. The line numbers refer to the track-change version.

This paper uses two years of topsoil temperature, snow, and active layer data from six vegetation types within a heterogeneous Low Arctic landscape in the northwestern Canadian Arctic to evaluate the relationships among these variables – vegetation, snow, soil temperature, and active layer depth. With changes in arctic vegetation being readily observed, there needs to be a greater understanding of how vegetation influences snow dynamics, ground temperature, and ultimately active layer depth and permafrost. Several papers in the literature do exist on this topic, however, the results collectively are not incredibly clear and consistent, and more data and analyses are needed. With a soil temperature dataset as robust as the one from this study, there are unlimited ways to analyze the data, and everyone will have their own opinion on how best to do that. The analysis presented here is generally a fine one, and is informative, and therefore I will not give opinions on other ways in which the analysis could have been done, but rather will provide some constructive comments on the existing analysis.

The following are some general, more specific, and some minor editorial suggestions:

**1)** State the depth of the data loggers in the Abstract We added the information to the abstract (l. 6).

**2) Line 28** Low Arctic is commonly capitalized, and change the "an" to "a" We removed the sentence in question for other reasons and corrected other occurrences of Low Arctic (l. 444, 449).

**3) Line 37** You mention that tussock tundra is commonly found in depressions, which is not always my experience, and I'm not sure how widespread that it. Certainly not on the exposed hilltops, but more often I think that tussock tundra is found on mesic slopes, as opposed to more saturated lowland positions. We agree with the reviewer that tussock tundra can often be found on mesic slopes and added this information to the introduction (l. 48). At our site specifically, 95% of the tussock patches have a slope of less than $4°$, which is by far the lowest value of all vegetation types (l. 116–117).

**4) Line 40** should be "active layer thickness at the end of summer" Changed (l. 51)

**5) Line 88** "gasses" should be "grasses"  Changed (l. 106)

**6) Lines 139-140** since you use "summer" here, I was wondering if you should define your seasons first. Also wondering if there is a better way of stating this – e.g. average topsoil temperature was more than 5 degrees C lower than air temperature.  We changed the order of the paragraphs to first define the seasons before we use this definition. We also replaced the sentence as suggested (l. 159–165, 168).

**7) Lines 140-142** how were these values determined, i.e. the -5 and +7 degrees C differences?  We decided to takes these thresholds based on the histogram in Figure A1. In case (b) (the average summer topsoil temperature was more than $5\,°C$ colder than air temperature) the three excluded time series where really a lot different from all the others and all thresholds between -5 and $-8\,°C$ had the same effect. In case (c) (more than 5% of the single summer measurements were more than $7\,°C$ above air temperature), the definition of the threshold was less clear and more subjective, also based on a close inspection of the time series. Other values would be possible but we think that we excluded the timeseries which were most prone to errors in this way.

**8) Line 148** Assuming that you smoothed the data prior to defining the seasons? Did smoothing remove all spikes in the data such that the winter and summer temperature thresholds yielded continuous seasons, i.e. were there any days within winter or summer that fell outside of the threshold temperatures?  We did not do any additional smoothing despite the two steps mentioned: (1) average all (365) daily air temperatures for 20 years and (2) smooth using a 7-day moving window. As you can see in Figure A2, this removed most spikes and the resulting seasons were continuous. I suppose in many environments such a simple approach would not yield continuous seasons. In our case, the temperature rise between winter and summer and the temperature drop end of summer until winter are very steep. Therefore, the definition of the seasons is not very sensitive to the defined temperature thresholds. We found that seasons defined in this way suit the data and environment much more than the classical 3-month per season. Especially the winter is clearly longer than December to February.

**9) Lines 153-154** I understand how the slopes for fall and spring might work directionally as cooling and warming respectively, but I would imagine that the peak warm and cool days with fall somewhere in the middles of summer and winter respectively, and therefore the slopes would not be very informative for these seasons.  Even if only a few cm below the soil surface, topsoil temperature generally lags behind air temperature, especially in winter under snow. Thus the coldest topsoil temperatures were observed in March, at the end of winter and the rate of cooling in winter is low but consistent and (we find) interesting. We agree that in summer (especially in the 2nd period), the warmest topsoil temperatures are not at the end of the season and the slope is less indicative and in some cases negative (when August is cold). We included the summer slope for consistency and prefer to keep it. However, we added an additional comment on negative summer slope values (l. 297–298).

**10) Lines 157-158** Would the day when temperatures first drop below 0.5 degrees C be the beginning of the freezing period in autumn?  This should be the case. However, we did not observe a clear beginning of the freezing period as all timeseries

showed significant peaks (warm periods) after the first cold temperatures (Figure 5ab). The freezeback was not very clear

60      for most sensors.

**11) Line 168** remove the comma after "those" Done (l. 199)

**12) Line 181** by "date of thawing" do you mean first date, end date, or both? The statement is true for both, start and end of thawing. We were mostly interested in the end of the thawing period (as in the sentence just before) and specified this now (l. 214).

65      **13) Line 199** give the actual temperature ranges The actual mean October temperatures were $-5.3\,^{\circ}\text{C}$ and $-1.0\,^{\circ}\text{C}$ for the coldest and warmest location in the first period and $-2.8\,^{\circ}\text{C}$ and $0.2\,^{\circ}\text{C}$ in the second period. We added these numbers to the manuscript (l. 234–236).

**14) Lines 263-264** with regard to the relationship between October soil temperatures and active layer depth, I have a hard time believing that Oct. temperatures can influence active layer depth – more likely the other way around maybe? Absolutely,

70      we agree. We replaced the misleading sentence (l. 325).

**15) Figure 8** I'm not sure that this figure is very useful. If vegetation is influencing snow depth, than it's very likely that snow depth in one year will be related to snow depth the next year. You might considering removing this. We agree that the former Figure 8 was not essential and moved it to the appendix (now Figure B2). We prefer to keep it in the manuscript because it supports our statement on wind redistribution in Section 3.2. Furthermore, the figure illustrates that the any

75      model snow depth based on vegetation type will be limited by the substantial interannual variability.

**16) Figure 7** you discuss these relationships a lot, but I don't see any statistical analyses on them (with the exception of panels (a) and (f)). Are these significant relationships across and within vegetation types? Also, if these relationships are considered to be causal, then snow depth should be on the x-axis, as it is driving the other variables (again, except panels (a) and (f)). These relationships are all statistically significant (p<0.01) and we added regression lines and correlation

80      coefficients to Figure 7 for some (basic) statistical analysis. We also swapped the figure axis because we agree that snow depth is rather the driver than the response.

**17) Figure 11** same as Figure 7 with regard to the statistical analyses (except panel (a)). Axes are fine in this figure as active layer depth is the assumed dependent variable. We added regression lines to Figure 10 (previously 11) in the case of significant (p<0.05) relationships and correlation coefficients to the figure for some (basic) statistical analysis. As

85      we discuss the differential relationships for different vegetation types in Section 4.4, we added additional panels with regression lines for lichen and dwarf shrubs and all other vegetation types combined.

**18) Line 327** remove the word "temperature." Done (l. -410)

**19) Lines 332-334** examples of where relationships are being discussed with no statistical analyses The reviewer is right, we did not show statistical analysis within single vegetation types although we discuss them (in particular for winter

90      temperature – ALT, snow melt timing – ALT and positive degree days – ALT). We added the statistical analysis of these relationships for two groups of vegetation types: lichen and dwarf shrubs versus taller vegetation to the results Section 3.4 (l. 310–324) and Figure 10 (previously 11). We also expanded the discussion on this topic in Section 4.4 (l. 410–421).

**20)** Finally, with regard to the Discussion and Conclusions, the vegetation may indeed have an effect on active layer depth,
95      through its effect on snowpack, as vegetation is effecting the snowpack, which is driving snowmelt and spring/summer temperature regimes. We agree that vegetation has an effect on active layer thickness or, at least, there is a correlation. As we describe in the results Section 3.4, vegetation explains 34% of the ALT variability (Table 2). We describe this effect as 'weak' even though it is statistically significant. We agree that this correlation is likely due to winter snow effects and added an additional comment to the discussion (l. 436).

[revised manuscript text omitted]